# Association between the donor to recipient ICG-PDR variation rate and the functional recovery of the graft after orthotopic liver transplantation: A case series

Vittorio Cherchi[1], Luigi Vetrugno[2,3], Giovanni Terrosu[1,3], Victor Zanini[2,3], Marco Ventin[3]*, Riccardo Pravisani[1,3], Francesco Tumminelli[1,3], Pier Paolo Brollo[1,3], Erica Boscolo[1,3], Roberto Peressutti[3], Dario Lorenzin[1], Tiziana Bove[2,3], Andrea Risaliti[3], Umberto Baccarani[1,3]

**1** General Surgery Clinic and Liver Transplant Center, University-Hospital of Udine, Udine, Italy,
**2** Department of Anesthesia and Intensive Care, University-Hospital of Udine, Udine, Italy, **3** Department of Medicine, University of Udine, Udine, Italy

* marco.ventin@hotmail.it

**Data Availability Statement:** All relevant data are within the paper and its Supporting Information files.

## Abstract

### Background

Despite current advances in liver transplant surgery, post-operative early allograft dysfunction still complicates the patient prognosis and graft survival. The transition from the donor has not been yet fully understood, and no study quantifies if and how the liver function changes through its transfer to the recipient. The indocyanine green dye plasma disappearance rate (ICG-PDR) is a simple validated tool of liver function assessment. The variation rate between the donor and recipient ICG-PDR still needs to be investigated.

### Materials and methods

Single-center retrospective study. ICG-PDR determinations were performed before graft retrieval (T1) and 24 hours after transplant (T2). The ICG-PDR relative variation rate between T1 and T2 was calculated to assess the graft function and suffering/recovering. Matched data were compared with the MEAF model of graft dysfunction.

### Objective

To investigate whether the variation rate between the donor ICG-PDR value and the recipient ICG-PDR measurement on first postoperative day ($POD_1$) can be associated with the MEAF score.

### Results

36 ICG-PDR measurements between 18 donors and 18 graft recipients were performed. The mean donor ICG-PDR was 22.64 (SD 6.35), and the mean receiver's ICG-PDR on 1st POD was 17.68 (SD 6.60), with a mean MEAF value of 4.51 (SD 1.23). Pearson's test

**Funding:** The author(s) received no specific funding for this work.

**Competing interests:** The authors have declared that no competing interests exist.

stressed a good, linear inverse correlation between the ICG-PDR relative variation and the MEAF values, correlation coefficient -0.580 (p = 0.012).

## Conclusion

The direct correlation between the donor to recipient ICG-PDR variation rate and MEAF was found. Measurements at T1 and T2 showed an up- or downtrend of the graft performance that reflect the MEAF values.

## Introduction

Despite 40 years of experience in the field, OLTx represents a challenging surgery and less than complication-free. With the translation of the graft from the donor to the recipient, the transplantation process represents a susceptible moment to numerous insults that can damage the organ with important repercussions on the graft and patient survival [1, 2]. In the immediate postoperative period, the graft dysfunction can become a serious complication [3, 4]. The prompt recognition of this condition is crucial to improve patient survival [5]. The Model of Early Allograft Function (MEAF) grades the severity of liver graft dysfunction by a continuous score, based on bilirubin, international normalized ratio, and alanine aminotransferase within three days post-transplant, and it is a prognostic tool for 3, 6 and 12-month patient and graft survival [6, 7]. Indocyanine green (ICG) is a cyanine dye safely used in medical diagnostics; although with different techniques, ICG boasts broad applications from vascular and lymphatic perfusion assessments in the fields of gastrointestinal surgery and gynecology, to the vascular and organ functional assessment in hepatology [8–10].

Assessment of the indocyanine green dye plasma disappearance rate (ICG-PDR) is a noninvasive technique that could be adopted as a point-of-care test (POCT) to evaluate the liver graft dysfunction bedside. The preliminary assessment of the graft functionality is essential to define its suitability for transplant. Traditionally, this judgement occurs through the evaluation of liver function tests and liver biopsy before or during organ retrieval [11].

Studies reported different ICG-PDR cut-off values in the recipient to have a prognostic role for graft dysfunction after OLTx [5, 12–14]. Clinical interest in this technology has varied greatly over the last twenty years, depending largely on the beliefs of the various transplant centres, and this exposes patient outcome to Bayesian probability, shifting it into a "zone of uncertainty" [15, 16]. Recently, some authors have demonstrated the similar predictive role of the ICG-PDR performed on the liver donor compared to early allograft dysfunction (EAD), without testing the assay on the recipient [17].

The transition from the donor to the recipient has not been yet fully understood, and no study quantifies how and if the liver function changes in its transfer to the recipient by applying the assay to study this transition. The hypothesis of our study is to analyze whether the variation rate between the donor ICG-PDR value and the recipient ICG-PDR measurement in the first postoperative day ($POD_1$) can be associated with the MEAF score and the patient's prognosis.

## Materials and methods

### Study design

This was a single center retrospective study analyzing the ICG-PDR variation rate between liver transplant donors immediately before harvesting and recipients at post-operative day 1

(POD$_1$). All organs were sourced from donors after brain death (DBD) hospitalized in the Intensive Care Unit of the University Hospital of Udine. The informed consent for the donation of organs was obtained with respect to the Italian law, art. 91/99 and 578/93. All transplantation procedures were performed at the University Hospital of Udine from October 2019 to December 2020. The study was conducted following the ethical principles deriving from the Declaration of Helsinki and was approved by our Institutional Review Board number 021/2021. None of the transplant donors was from a vulnerable population. The patient's willing to participate in clinical studies was obtained through the general consent (GECO) system, and the European General Data Protection Regulation 2016/679 was (GDPR) respected. A general informed consent for research purposes was signed by all admitted patients and the same standard-of-care was applied.

## Study population

All liver grafts were harvested from donors after brain death (DBD). Only elective patients with end-stage liver disease scheduled through the Model for End-stage Liver Disease (MELD) score were enrolled in this study. The inclusion criteria were: (a) ICG-PDR testing performed on the donor just before liver procurement and (b) ICG-PDR testing performed on the first postoperative day (POD) on respective graft recipients. Exclusion criteria were: intra-operative mortality, acute vascular complications, active bleeding in the immediate postoperative period (24 hours) and known iodine allergy. The peri- and postoperative collected data included details regarding donor's and recipient's ICG-PDR values, demographical data, graft characteristics, blood transfusion requirements during surgery and blood chemistry in the first seven days after OLTx.

## Study protocol

ICG-PDR determinations were tested in the intensive care unit before entering the operating room for graft retrieval (T1) and 24 hours after OLTx surgery (T2). ICG-PDR was assessed by pulse spectrophotometry to provide a real-time picture of a transplanted graft's metabolic function before and after surgery. A detailed explanation of ICG-PDR measurement is reported in our previous publication [18]. To estimate the graft function and recovering/suffering ICG-PDR relative variation rate between T1 and T2 was calculated. These data were subsequently matched and analyzed with graft dysfunction described as MEAF. According to the literature, the MEAF score was assessed for each recipient considering the highest INR, ALT and total bilirubin within the first three postoperative days:

$$\text{MEAF} = (\text{score ALT}_{max3POD} + \text{score INR}_{max3POD} + \text{score bilirubin}_{3POD})$$
$$\text{score ALT}_{max3POD} = 3.29 / [1 + e^{-1.9132} (\ln (\text{ALT}_{max3POD}) - 6.1723)]$$
$$\text{score INR}_{max3POD} = 3.29 / [1 + e^{-6.8204} (\ln (\text{INR}_{max3POD}) - 0.6658)]$$
$$\text{score bilirubin}_{3POD} = 3.4 / [1 + e^{-1.8005} (\ln (\text{bilirubin}_{3POD}) - 1.0607)]$$

The MEAF Scoring System provides a significant association with patient survival for the 3-, 6-, and 12-month follow-up, including that the higher the MEAF Score is, the lower the recipients' and grafts' survival rates are [6].

All patients were considered with a minimum follow-up of 90 days. The ICG-PDR log-variation rate was associated with a new composite endpoint called Hospital-Free Days (HFD). HFD summarizes the postoperative outcomes into the number of days patients spend outside of any healthcare facility in an observation period of 90 days after OLTx. ICG-PDR testing on liver transplant recipients has been a standard-of-care for 20 years at our institution. From 2019, ICG-PDR testing has been implemented as standard-of-care for all (DBD) liver donors and transplant recipients. Since the feasibility of performing ICG-PDR tests in the donor

before entering the operating room for graft retrieval may not be equally accepted worldwide, ethical and legal regulations from each country may limit the applicability of the findings from this study.

## Outcome measures

The primary study outcome was to evaluate the association between the distribution of the relative variation of the ICG-PDR values before and after surgery (T1 and T2) and the MEAF score.

## Statistical analysis

Considering the results of the Shapiro-Wilk test for normality, quantitative variables were presented as means and standard or median deviations plus the range. Qualitative variables are expressed as absolute and relative frequencies. We assessed the distribution of the relative variation in ICG-PDR values [(logICG-PDR2 –logICG-PDR1)/ICG-PDR1] at the two different time points (T1 before explant, and T2 at the first post-OLTx day). We applied Pearson's test to assess for statistically significant correlation with MEAF. Subsequently, the linear regression analysis was performed. We tested via ANOVA the presence of any association between MEAF and dichotomic variables or via correlation test if the analyzed variable was continuous. The association between qualitative variables was tested using the $\chi 2$ chi-square test and Fisher's exact test. Statistical significance was considered for p-values $\leq 0.05$. MedCalc Statistical Software version 16.4.3 (MedCalc Software bv, Ostend, Belgium; 2016) from the text box into the manuscript was used.

## Results

During the study period 36 ICG-PDR measurements between 18 donors and 18 graft recipients were performed. Demographical, clinical and patient history data are shown in Tables 1 and 2. The mean age of donors and recipients was 51.5 and 56.3 years, respectively (SD 16.25 and 10.48), mean DRI was 1.46 (SD 0.34), mean recipient BMI was 24.36 (SD 3.12), mean MELD-Score was 12.5 (IQE 10.0–17.0). The mean donor ICG-PDR was 22.64 (SD 6.35), and the mean receiver's ICG-PDR on 1st POD was 17.68 (SD 6.60), mean MEAF value was 4.51 (SD 1.23), Table 3. ICG-PDR values show a tendency to decrease between time zero (T1) and 24 h post-OLTx (T2) in the cohort of patients who presented with higher MEAF values. The assessment of the distribution of the relative variation in ICG-PDR was calculated at two times, as mentioned above. We applied Pearson's test that underlined a good, linear inverse correlation between the relative variation of the ICG-PDR and the MEAF values with a correlation coefficient of -0.580 (p = 0.012). Figs 1 and 2; Table 4. Linear regression analysis discovered an $R^2$ value of 0.336. HFD after OLTx, showed a mean value of 64.78 (SD 6.71) days. A mean of 68.67 (SD 6.56) days was notable for patients showing a positive up-trending log ICG-PDR variation rate, vs 62.83 (SD 6.13) days within the transplants showing a negative ICG-PDR variation rate, although without reaching statistical significance. The ICG-PDR value at T2 stressed the correlation with MEAF, (p = 0.002). No deaths, no PNF and no re-transplants were observed in a 90-day follow-up period.

## Discussion

The main findings of our study are as follows: first, we found an association between the variation rate of the donor-recipient ICG-PDR and MEAF; secondly, we found that the ICG-PDR on the first postoperative day was related with MEAF. The organ transition from the donor to

**Table 1. Donor demographical and laboratory data.**

| | |
|---|---|
| Total number of patients | 18 |
| Age, years (mean, SD) | 51.50 (16.25) |
| Male/Female | 10 / 8 |
| BMI kg/m$^2$ (mean, SD) | 26.38 (5.27) |
| **Biochemical values** | |
| Lactates, mEq/L (mean, SD) | 1.07 (0.90–1.21) |
| Bilirubin, mg/dL (mean, SD) | 2.48 (0.99) |
| ALT, IU (mean, SD) | 61.37 (8.22) |
| AST, IU (mean, SD) | 56.05 (8.40) |
| AP, IU (mean, SD) | 55.39 (38.57) |
| GGT, IU (median, IQR) | 23.5 (14.0–33.0) |
| Creatinine, mg/dL (mean, SD) | 1.06 (0.38) |
| INR, % (median, IQR) | 1.08 (1.02–1.27) |
| Sodium, mEq/L (mean, SD) | 150.94 (9.37) |
| **Graft characteristics** | |
| ICG-PDR, %/min T1 (mean, SD) | 22.64 (6.35) |
| Steatosis, % (mean, SD) | 12 (8.93) |
| Donor Risk Index, n (mean, SD) | 1.46 (0.34) |
| Cold Ischemia time, min (mean, SD) | 372.89 (91.36) |
| Warm Ischemia time, min (median, IQR) | 35 (30–40) |
| **Cause of Death** | |
| Traumatic head injury, n (%) | 5 (27.8) |
| Haemorrhagic stroke, n (%) | 13 (72.3) |

BMI, Body Mass Index; ALT, Alanine AminoTransferase; AST, Aspartate Amino-Transferase; AP, Alkaline Phosphatase; GGT, gamma-glutamyl transferase; INR, International Normalized Ratio; ICG-PDR, IndoCyanine Green dye Plasma Disappearance Rate; IQR, Interquartile Range; SD, Standard Deviation; n, number; min, minutes

**Table 2. Recipient demographical and laboratory data.**

| | |
|---|---|
| Total number of Patients | 18 |
| Age, years (mean, SD) | 56.28 (10.48) |
| Male/Female | 11 / 7 |
| BMI kg/m$^2$ (mean, SD) | 24.36 (3.12) |
| MELD score, n (median, IQR) | 12.5 (10–17) |
| LOS, days (median, IQR) | 20 (17–26) |
| **Biochemical values on 1$^{st}$ PO day** | |
| SOFA score on 1$^{st}$ PO, n (median, IQR) | 7 (6–7) |
| Lactates on 1$^{st}$ POD, mEq/L (mean, SD) | 1.11 (0.48) |
| Bilirubin on 1$^{st}$ POD, mg/dL (median, IQR) | 3.05 (1.81–6.97) |
| ALT on 1$^{st}$ PO, IU (median, IQR) | 695.5 (502.0–1205.0) |
| AST on 1$^{st}$ PO, IU (median, IQR) | 809.5 (465.0–1951.0) |
| Creatinine, mg/dL (mean, SD) | 1.35 (0.57) |
| INR, % (mean, SD) | 1.47 (0.20) |
| **Graft characteristics on 1$^{st}$ PO day** | |
| ICG-PDR, %/min T2 (mean, SD) | 17.68 (6.60) |

BMI, Body Mass Index; MELD, Model for End-stage Liver Disease; IU, International Units; SOFA score, Sequential Organ Failure Assessment score; ALT, Alanine AminoTransferase; AST, Aspartate Amino-Transferase; INR, International Normalized Ratio; ICG-PDR, IndoCyanine Green dye Plasma Disappearance Rate; IQR, Interquartile Range; SD, Standard Deviation; n, number; min, minutes; 1$^{st}$ POD, first Post-Operative Day

**Table 3. Outcomes.**

| Outcomes | |
|---|---|
| MEAF (mean, SD) | 4.51 (1.23) |
| HFD, days (mean, SD) | 64.78 (6.71) |
| HFD [in positive logICG-PDR var rate (mean, SD)], days (mean, SD) | 68.67 (6.56) |
| HFD [in negative logICG_-PDR var rate (mean, SD)], days (mean, SD) | 62.83 (6.13) |
| ICG-PDR variation rate (mean, SD) | -0.0048 (0.0089) |
| PNF, n (%) | 0 (0) |
| Re-OLTx, n (%) | 0 (0) |
| Deaths [90 –day mortality], n (%) | 0 (0) |

MEAF, Model for Early Allograft Function; HFD, Hospital-Free Days; ICG-PDR variation rate, IndoCyanine Green dye Plasma Disappearance Rate variation rate calculated as follows [(logICG-PDRt2 –logICG-PDRt1)/ICG-PDRt1]; PNF, Primary Non-Function; Re-OLTx, Re–transplantation; IQR, Interquartile Range; SD, Standard Deviation; n, number; min, minutes; 1st PO, first Post-Operative Day.

the recipient is very sensitive to several insults that can lead to the impaired postoperative functional recovery of the graft. Liver transplantation represents the final stage of a complex multi-phase procedure consisting of organ procurement, preservation, preparation, and implantation. Ischemia-reperfusion damage (IRI), the quality of the donor graft, and the recipient's condition at the time of transplantation can be the main factors responsible for the onset of graft dysfunction [19]. The overall quality of the graft depends on the donor's age, the degree of steatosis (assessed by liver biopsy) or the use of marginal donor organs from "Extended criteria donors" (ECD). Given the growing demand for organs, grafts from ECD

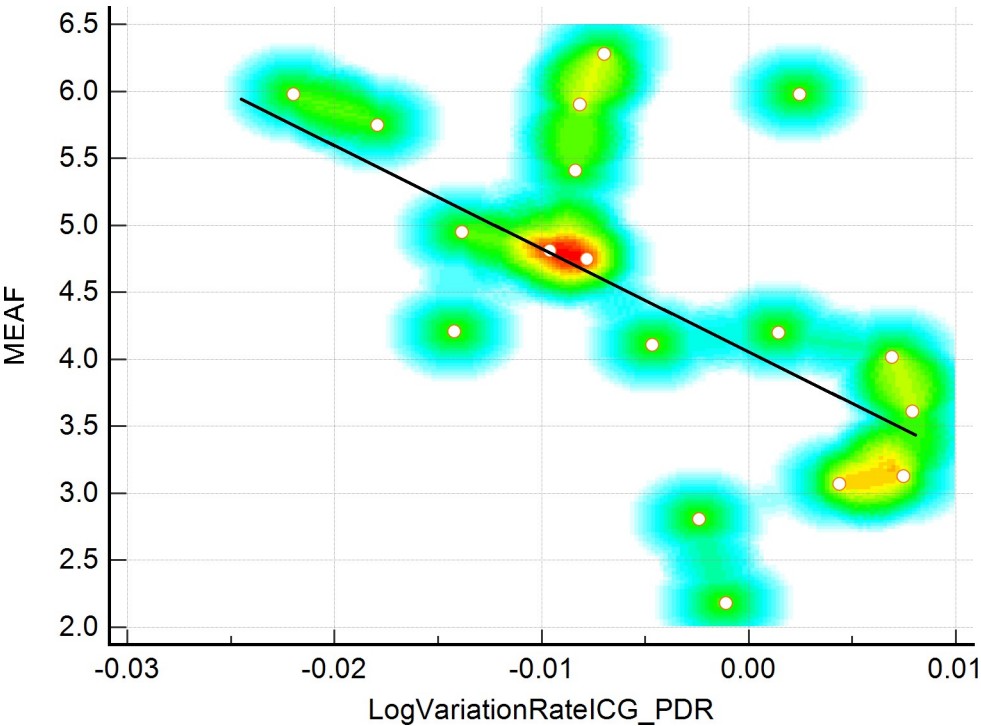

**Fig 1. Estimation plot for paired ICG-PDR samples.**

# Estimation Plot

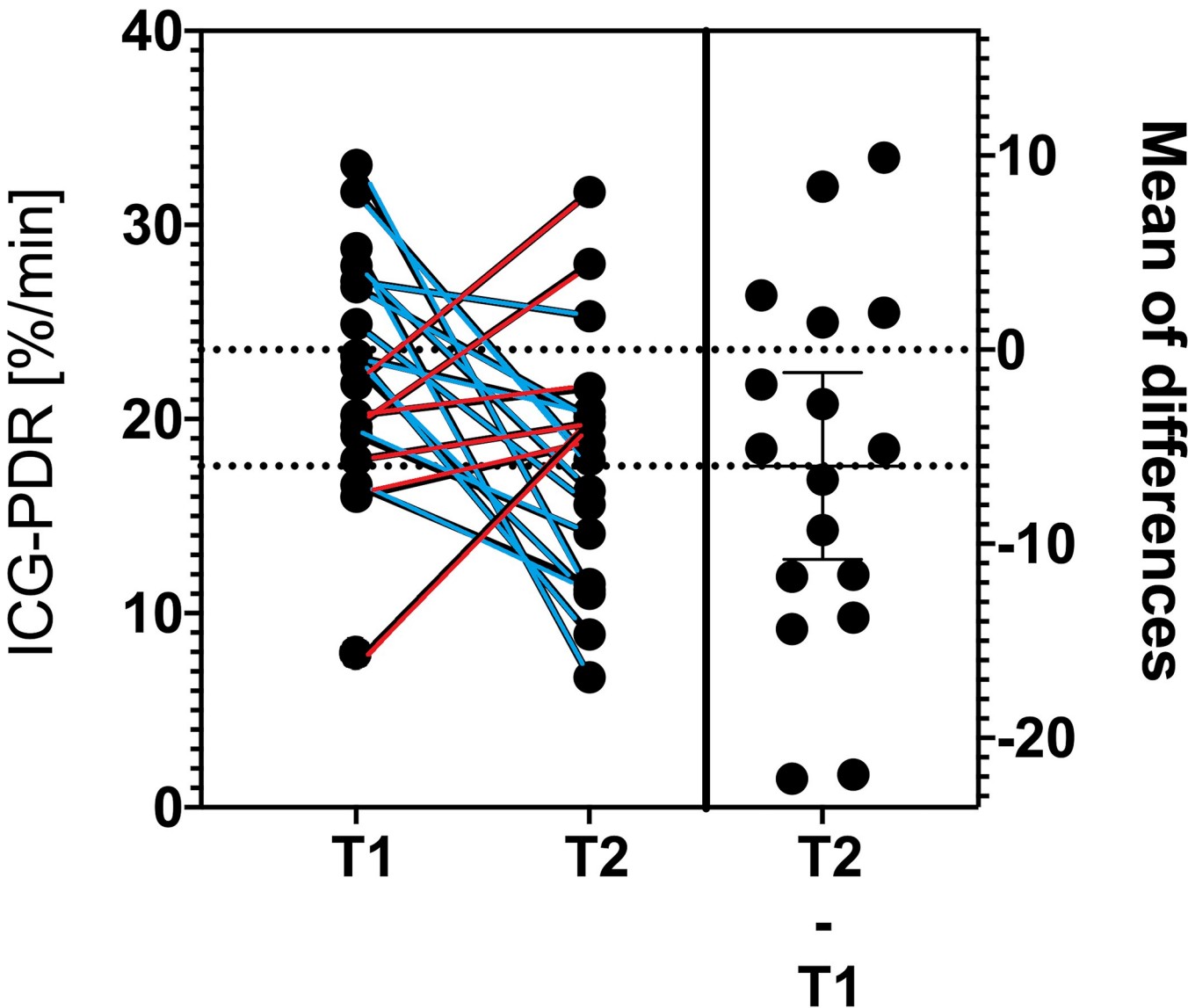

**Fig 2. Scattered comparison graph.**

constitute an indispensable resource. The prompt recognition of the donor liver function is of paramount importance to predict the postoperative liver-graft function and to improve the patient survival. The MEAF score, as literature reports, should be considered the gold standard to predict and detect the occurrence of graft dysfunction, with its intrinsic limitations as either moderately "good" or moderately "bad" [7].

**Table 4. Association between variables and the distribution of the relative variation in ICG-PDR.**

| | Distribution of the relative variation in ICG-PDR, significance level expressed as p-value |
|---|---|
| Age of recipient | p-value = 0.357 |
| BMI recipient | p-value = 0.853 |
| Cold ischemia time | p-value = 0.236 |
| Warm ischemia time | p-value = 0.161 |
| UEC | p-value = 0.792 |
| Plasma | p-value = 0.235 |
| MEAF | p-value = 0.012 |
| Lactates on 1st POD | p-value = 0.378 |
| AST on 1st PO | p-value = 0.272 |
| ALT on 1st PO | p-value = 0.376 |
| Serum albumin on 1st POD | p-value = 0.016 |
| GGT on 1st POD | p-value = 0.143 |
| HFD | p-value = 0.079 |
| Age of donor | p-value = 0.103 |
| BMI donor | p-value = 0.957 |
| Steatosis | p-value = 0.617 |
| Donor risk index | p-value = 0.141 |
| Lactates of donor | p-value = 0.102 |
| AST of donor | p-value = 0.305 |
| ALT of donor | p-value = 0.414 |
| GGT of donor | p-value = 0.017 |
| MELD | p-value = 0.104 |

MELD, Model for End-stage Liver Disease; ICG-PDR, IndoCyanine Green dye Plasma Disappearance Rate; UEC N°, Number of Units of packed red blood cells; ALT, Alanine AminoTransferase; AST, Aspartate Amino-Transferase; GGT, Gamma Glutammil-Tranferase: IQR, Interquartile Range; HFD, Hospital Free Days; MEAF, Model for Early Allograft Dysfunction.

Over the last decades, indocyanine green plasma disappearance rate (ICG-PDR) aroused modest interest between researchers and clinicians as a tool of liver function assessment. Numerous studies about the efficacy of the ICG-PDR testing performed on the first postoperative day to predict the graft performance and early outcome after OLTx have been published [20, 21]. Some authors observed an association between low ICG-PDR values and the onset of EAD after transplantation with a decrease in overall survival [5]. From our previous study based on a cohort of 78 patients, the ICG-PDR on the 1st PO day represents an easy, repeatable, and bedside measurement able to predict EAD and graft survival at 1- and 5-years with a cut-off value of 16%/min [18]. Other authors observed that low ICG-PDR values, respectively < 12.85 and < 9.6%/min, were associated with early post-operative complications and death [5, 13, 22]. In 2014, Klinzing et al. observed that the combination of a high recipient MELD score (MELD >25) and a low ICG-PDR measured in the early postoperative phase (ICG-PDR < 20%/minute) was significantly related to longer supportive intensive care stay and hospitalization for liver transplant recipients [23].

Few studies have focused their attention on testing ICG-PDR on liver donors. Koneru et al. (1994) pioneered the ICG-PDR for the quality assessment of the graft. The authors showed that an ICG-PDR value lower than 15%/min was an objective evaluation to predict transplant

graft quality [24]. Vos et al. (2014) studied the ICG-PDR to assess graft suitability from cadaveric donors before the organ retrieval. The authors found the ICG-PDR value <15%/min during the donor observation period to be associated with a poor outcome of the graft after transplantation [25]. Zarrinpar et al. (2015) considered the ICG-PDR testing to assess donors after brain death (DBD) and observed a relationship with graft survival [26]. In 2017, Tang et al. confirmed that the ICG-PDR testing in donors after brain death could be an effective quantitative method to predict functional graft recovery and improve early graft prognosis after liver transplantation [27]. These findings suggest that poor quality grafts are more sensitive to ischemia-reperfusion injury than the group with a lower donor risk index (DRI). More recently, in 2020, JM Asencio et al. studied the ICG-PDR from DBD donors to predict graft rejection in the recipients [28].

The low cost of the indocyanine green test, its simplicity, and the short time it took to perform it on both donors and recipients made it a valid tool to assess the progressive status of the graft function before and after transplantation.

In our study, we analyzed the difference between two measurements of ICG-PDR for the same graft before and after liver transplantation. We calculated the variation rate in ICG-PDR (ICG-PDR-rate) as an expression of graft damage, and we found that the variation is directly correlated with the Model of Early Allograft Function (MEAF). To our knowledge, before this study, there was no research in the literature investigating the ICG-PDR in both donor and recipient livers.

The calculated ICG-PDR variation rate is a dynamic index that assesses over time the evolution of the liver metabolic balance from the donor to the recipient and opens the way to numerous in-depth studies.

From our analysis, we noted that in two thirds of the cohort the ICG-PDR variation rate decreased, suggesting a worsening in the graft-liver function. Higher MEAF scores among these patients also coherently explained the trend. Interestingly, the ICG-PDR variation rate increased between the rest of the population which was also notable for lower MEAF scores. ICG-PDR measurement performed at T1 (donor) and T2 (recipient) showed an up- or downtrend of the graft performance, able to reflect the MEAF values. Additional data from the donor, such as, graft steatosis, warm and cold ischemia times, did not correlate with MEAF in our case series. This finding can probably rely on the condition of the donor himself.

The hypothesis that ICG-PDR variation rate could represent a measure of the ischemia and reperfusion injury (IRI) as previously hinted by Plevris et al seems to be supported by the results of our study showing good correlation between ICG-PDR values from donors to recipients and MEAF scores post-transplant [29].

The intention of photographing the liver function in two different moments of the transplant process, revealed an interesting finding. If we look at the daily practice, it is arguable to assess the graft performance simply and objectively after OLTx. The ICG-PDR variation rate can be seen as a simple tool to appraise the early outcomes and could open an interesting window for the application of this technique to in the daily routine of the transplant surgeons. In fact, ICG-PDR is a simple and cost-effective tool, simple to read and above all simple to compare.

Furthermore, this is a pilot-study and some limitations also need to be acknowledged: we reviewed our database retrospectively. We did not perform any power analysis about the study's primary aim but only analyzed the available data. In patients with low cardiac output or massive active bleeding the ICG-PDR may not provide reliable information, therefore ICG-PDR was performed only in the case of hemodynamic stability. Moreover, the small cohort of patients did not allow us to investigate certain aspects in depth. It would be interesting to deepen the current knowledge about the increase/decrease in donor-recipient PDR

observed in some cases. Certainly, by expanding the sample size, it will be possible to increase the variety of procured grafts and investigate the role of the PDR rate in such contexts.

## Conclusions

In conclusion, we found a direct correlation in ICG-PDR variation rate from donor-recipient and MEAF. ICG-PDR variation rate is an easy, bedside, repeatable measurement to estimate graft perioperative liver graft dysfunction. Further studies are needed to confirm our results in a large population and between centers. That will make it possible in the future to outline through ICG-PDR variation rate a tailored transplant pathway, aiming to contain postoperative complications.

## Supporting information

**S1 Data.**
(XLSX)

## Author Contributions

**Conceptualization:** Vittorio Cherchi, Luigi Vetrugno.

**Data curation:** Victor Zanini, Marco Ventin, Riccardo Pravisani, Francesco Tumminelli, Pier Paolo Brollo, Erica Boscolo.

**Formal analysis:** Victor Zanini, Riccardo Pravisani, Pier Paolo Brollo.

**Investigation:** Erica Boscolo, Dario Lorenzin.

**Methodology:** Luigi Vetrugno, Marco Ventin.

**Project administration:** Umberto Baccarani.

**Supervision:** Giovanni Terrosu, Roberto Peressutti, Dario Lorenzin, Tiziana Bove, Andrea Risaliti, Umberto Baccarani.

**Validation:** Andrea Risaliti, Umberto Baccarani.

**Writing – original draft:** Vittorio Cherchi, Luigi Vetrugno, Marco Ventin.

**Writing – review & editing:** Luigi Vetrugno, Victor Zanini, Marco Ventin, Francesco Tumminelli, Umberto Baccarani.

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
