## [Decision Letter · Decision Letter 0]

22 Jun 2021

PONE-D-21-16544

ASSOCIATION BETWEEN THE DONOR TO RECIPIENT ICG-PDR VARIATION RATE AND THE FUNCTIONAL RECOVERY OF THE GRAFT AFTER ORTHOTOPIC LIVER TRANSPLANTATION: A PILOT STUDY

PLOS ONE

Dear Dr. Ventin,

Thank you for submitting your manuscript to PLOS ONE. After careful consideration, we feel that it has merit but does not fully meet PLOS ONE’s publication criteria as it currently stands. Therefore, we invite you to submit a revised version of the manuscript that addresses the points raised during the review process.

I have read with great interest the manuscript entitled ‘Association between the donor to recipient ICG-PDR variation rate and the functional recovery of the graft after orthotopic liver transplantation: a pilot study’. In this original article (single centre retrospective study), authors found an association between the variation rate of the donor-recipient ICG-PDR and MEAF; and, also, an association between the ICG-PDR on the first postoperative day and MEAF. The manuscript is written well and the topic of clinical interest. As stated by the authors, indeed, identifying efficient predictors of postoperative liver graft function is crucial because of the expansion of extended criteria donor graft utilisation.

Thirty-six ICG-PDR measurements were performed for 18 donors and 18 graft recipients. Only donors after brain death (DBD) were included. In addition, the reported features of these donors suggest they are good quality organs (e.g., good donor ICG-PDR, %/min T1 (mean, SD) 22.64 (6.35)). Recipients were predominantly low risk cases (high ICG-PDR, %/min T2 (mean, SD) 17.68 (6.60)/ MELD Score was 12.5 (IQE 10.0-17.0); mean CIT 383.50 minutes (SD 91.36)).

The results reported are original and have not been published elsewhere. The study was conducted according to ethical standards.

Although experiments and statistics were performed to a high technical standard and are described in sufficient detail, the experienced reviewers raised some questions regarding sample size and the outcomes analysed. They have also asked for further information on the aim of applying ICG-PDR in this context and the time point to measure the MEAF score.

I echo reviewer 1, and if ` ICG-PDR testing on liver transplant recipients has been a standard-of-care for 20 years at our institution`, why not increase the donor pool? In addition, if ` All patients were considered with a minimum follow-up of 90 days`, why not patient and graft survival at 90-day?

Furthermore, indeed, the inclusion of data on `hard` endpoints would increase the quality of the study. Consider report patient and graft survival and the occurrence of early allograft dysfunction by conventional criteria (as those proposed by Olthoff et al.). Those would be important considering the intention to propose a predictive test of graft performance and early outcome after transplantation.

In the Discussion, the authors advocate that the current study supports the hypothesis that ICG-PDR variation rate could represent a measure of the ischemia-reperfusion injury. However, they must admit that limitations of the study population (good quality grafts and low-risk recipients) may compromise this assumption.

Also, in the Discussion, authors must comment on the feasibility of performing ICG-PDR tests in the donor before entering the operating room for graft retrieval. In many countries, such interventions are not allowed in the donor. Therefore, ethical and legal regulations must limit the applicability of these findings.

To sum up, major concerns were raised by reviewers regarding methodological features, interpretation of results, and how they can affect the conclusions.

Authors must address all these points, consider those constraints, and the Conclusion of the paper. Please provide a point-by-point response to reviewers’ comments.

We look forward to receiving your revised manuscript.

Kind regards,

Yuri Longatto Boteon, M.D., Ph.D.

Academic Editor

PLOS ONE

Journal Requirements:

2. We note that your study involved tissue/organ transplantation. Please provide the following information regarding tissue/organ donors for transplantation cases analyzed in your study.

     1. Please provide the source(s) of the transplanted tissue/organs used in the study, including the institution name and a non-identifying description of the donor(s).

     2. Please state in your response letter and ethics statement whether the transplant cases for this study involved any vulnerable populations; for example, tissue/organs from prisoners, subjects with reduced mental capacity due to illness or age, or minors.

     - If a vulnerable population was used, please describe the population, justify the decision to use tissue/organ donations from this group, and clearly describe what measures were taken in the informed consent procedure to assure protection of the vulnerable group and avoid coercion. 

     - If a vulnerable population was not used, please state in your ethics statement, “None of the transplant donors was from a vulnerable population and all donors or next of kin provided written informed consent that was freely given.”

     3. In the Methods, please provide detailed information about the procedure by which informed consent was obtained from organ/tissue donors or their next of kin. In addition, please provide a blank example of the form used to obtain consent from donors, and an English translation if the original is in a different language.

     4. Please indicate whether the donors were previously registered as organ donors. If tissues/organs were obtained from deceased donors or cadavers, please provide details as to the donors’ cause(s) of death.

     5. Please provide the participant recruitment dates and the period during which transplant procedures were done (as month and year). 

     6. Please discuss whether medical costs were covered or other cash payments were provided to the family of the donor. If so, please specify the value of this support (in local currency and equivalent to U.S. dollars).

Reviewers' comments:

Reviewer's Responses to Questions

**Comments to the Author**

1. Is the manuscript technically sound, and do the data support the conclusions?

Reviewer #1: Yes

Reviewer #2: Partly

2. Has the statistical analysis been performed appropriately and rigorously? 

Reviewer #1: Yes

Reviewer #2: No

3. Have the authors made all data underlying the findings in their manuscript fully available?

Reviewer #1: Yes

Reviewer #2: Yes

4. Is the manuscript presented in an intelligible fashion and written in standard English?

Reviewer #1: Yes

Reviewer #2: Yes

5. Review Comments to the Author

Reviewer #1: Cherchi et al. evaluated the variation of ICG-PDR in donor and recipient liver. They have found a significant correlation between MEAF value and ICG-PDR variation.

The paper is well-written and organized. Several papers have reported the usefulness of the ICG test of transplant liver to predict early outcomes. This study is unique in which they analyzed the transition of ICG of the same liver from the donor to the recipient. Here are several comments.

First of all, the cohort size is too small. To attract more readers, the authors should add more patients or prolong the observation period and evaluate the variation ratio with the actual patient outcome, not with the MEAF.

ICG is now used in other fields other than the hepatology field to evaluate vascularity or blood flow. Please mention this in the introduction section.

The result of ICG would be affected by cardiac function. Please include the limitation of ICG testing itself.

Since this is a retrospective study, outcomes of the LT will be available. Although the authors used MEAF as a surrogate marker for the outcome, is there any correlation with a real outcome?

T1 ICG was obtained from DBD, and the liver should be normal. Why was T1 high in some patients?

DO the authors think ICG-PDR is a real-time test to assess graft dysfunction or a predictive marker for future graft dysfunction?

Reviewer #2: Dear Author

Congratulations on your study.

I have some concerns about and I would like to discuss with you.

If you have since 2019, Icg-PDR implemented as standard-of-care for all livers donors and transplant recipientes why you don't try increased your cohort of patients? How many liver transplants do you perform in your center in a year? Maybe with this analysis will be possible have more consistent and definitive results.

You put MELD score in table 1. Donor demographical and laboratory data. Please consider to change to Table 2.

Why did you prefer to use the MEAF grade instead other score like the definition proposed by Olthoff et al. ? And why did you use the MEAF score at day one? instead day 3 ?

This could minimizes the contribution of preoperative elevated bilirubin and significant preoperative coagulopathy found in the recipients.

When you discuss data from donor like steatosis, warm and cold ischemic times, for example, you did not correlate with MEAF, only because the good condition of the donor himself ? Did you considered that could be because you performed the score to soon after liver transplant, on day 1? Or maybe because you had only patients with low MELD and good donors conditions? I would like to know your opinion about.

The value of ICG-PDR on he first day after transplant is a good predictor for EAD, especially to livers from extended criteria, but you did not use those kind of donors, so the variation rate of ICG "represent a measure of the ischemia and reperfusion injury", like you hypothetically said. So this was not related with donor qualities. Do you agree ?

You have not seen your data any early allograft dysfunction (EAD), you use only good quality donors and recipients with low MELD. Possibily if our data was more extensive you coud find diferente results? Please I would like to hear your opinion about.

Thank you.

6. PLOS authors have the option to publish the peer review history of their article (what does this mean?). If published, this will include your full peer review and any attached files.

Reviewer #1: No

Reviewer #2: No

---

## [Author Response · Author response to Decision Letter 0]

26 Jul 2021

To: Yuri Longatto Boteon, M.D., Ph.D.

 Academic Editor

 PLOS ONE 

Udine, 7th July 2021

Subject: Revision of the article ID: PONE-D-21-16544; ASSOCIATION BETWEEN THE DONOR TO RECIPIENT ICG-PDR VARIATION RATE AND THE FUNCTIONAL RECOVERY OF THE GRAFT AFTER ORTHOTOPIC LIVER TRANSPLANTATION: A PILOT STUDY 

Dear Editor,

Thank you for inviting us to submit a revised draft of the above mentioned manuscript. We appreciate the time and effort you and the reviewers have dedicated, providing insightful feedback on how to strengthen our paper. Thus, we are very pleased to resubmit the revised article for further consideration. Please find below our answers to your and to the reviewer’s comments. 

We have carefully revised our manuscript accordingly to your suggestions. We believe that the findings reported in this manuscript are important, interesting to the readers of PlosONE Journal and worthy of publication. We hope our revised manuscript is satisfactory. 

In closing, we hope that our modifications have addressed all the issures raised in your informative and thorough review of the paper. We would be glad to make further changes upon request. 

On behalf of all the authors, we extend our gratitude to you for your time and assistance with our review. 

We look forward to hearing your response. 

Sincerely yours, 

Marco Ventin, M.D., and Prof. Luigi Vetrugno, M.D.

 

Reply:

Dear Dr. Ventin, 

Thank you for submitting your manuscript to PLOS ONE. After careful consideration, we feel that it has merit but does not fully meet PLOS ONE’s publication criteria as it currently stands. Therefore, we invite you to submit a revised version of the manuscript that addresses the points raised during the review process. 

1) I have read with great interest the manuscript entitled ‘Association between the donor to recipient ICG-PDR variation rate and the functional recovery of the graft after orthotopic liver transplantation: a pilot study’. In this original article (single centre retrospective study), authors found an association between the variation rate of the donor-recipient ICG-PDR and MEAF; and, also, an association between the ICG-PDR on the first postoperative day and MEAF. The manuscript is written well and the topic of clinical interest. As stated by the authors, indeed, identifying efficient predictors of postoperative liver graft function is crucial because of the expansion of extended criteria donor graft utilisation.

Reply 1: Thank you!

2) Thirty-six ICG-PDR measurements were performed for 18 donors and 18 graft recipients. Only donors after brain death (DBD) were included. In addition, the reported features of these donors suggest they are good quality organs (e.g., good donor ICG-PDR, %/min T1 (mean, SD) 22.64 (6.35)). Recipients were predominantly low risk cases (high ICG-PDR, %/min T2 (mean, SD) 17.68 (6.60)/ MELD Score was 12.5 (IQE 10.0-17.0); mean CIT 383.50 minutes (SD 91.36)).

Reply 2: ICG-PDR measurement performed at T1 (donor) and T2 (recipient) showed an up- or downtrend of the graft performance, able to reflect the MEAF values. As highlited by the Editor, the ICG-PDR could suggest the good quality of the organs: however, no study has been performed to date, to assess the ICG-PDR basal values to define an organ as of “good” or “poor” quality. In this study our main purpose was to monitorate the transition of the graft from the donor to the recipient through the ICG-PDR and finally comparing it with the currently most reliable predictive score of graft and patient’s prognosis. Please see ref: (Jochmans I, Fieuws S, Monbaliu D, Pirenne J. "Model for Early Allograft Function" Outperforms "Early Allograft Dysfunction" as a Predictor of Transplant Survival. Transplantation. 2017 Aug;101(8):e258-e264. doi: 10.1097/TP.0000000000001833. PMID: 28557956.) To address this comment and that from reviewer #1 we have implemented table 1 presenting the donor characteristics. Please see the new version of the manuscript. 

3) The results reported are original and have not been published elsewhere. The study was conducted according to ethical standards.

Reply 3: Thank you!

4) Although experiments and statistics were performed to a high technical standard and are described in sufficient detail, the experienced reviewers raised some questions regarding sample size and the outcomes analysed. They have also asked for further information on the aim of applying ICG-PDR in this context and the time point to measure the MEAF score.

Reply 4: We thank you for your question because thanks to this comment we have realized that our content can be misunderstood. We chose the MEAF score since recent evidence points towards it outperforming EAD in terms of patient and graft survival after transplant. Please find the relative references below:

- Jochmans I, Fieuws S, Monbaliu D, Pirenne J. "Model for Early Allograft Function" Outperforms "Early Allograft Dysfunction" as a Predictor of Transplant Survival. Transplantation. 2017 Aug;101(8):e258-e264. doi: 10.1097/TP.0000000000001833. PMID: 28557956.

- Pareja E, Cortes M, Hervás D, Mir J, Valdivieso A, Castell JV, Lahoz A. A score model for the continuous grading of early allograft dysfunction severity. Liver Transpl. 2015 Jan;21(1):38-46. doi: 10.1002/lt.23990. Epub 2014 Nov 24. PMID: 25204890.

- Barrueco-Francioni JE, Seller-Pérez G, Lozano-Saéz R, Arias-Verdú MD, Quesada-García G, Herrera-Gutiérrez ME. Early graft dysfunction after liver transplant: Comparison of different diagnostic criteria in a single-center prospective cohort. Med Intensiva (Engl Ed). 2020 Apr;44(3):150-159. English, Spanish. doi: 10.1016/j.medin.2018.09.004. Epub 2018 Oct 24. PMID: 30528954. 

Being a continuous grading scale for graft dysfunction, the MEAF score can provide additional, granular information that could be used both clinically and as a surrogate endpoint of transplant survival in clinical trials unlike the categorical values obtained for EAD. Therefore, we found it statistically and methodologically more appropriate to study the potentienl relationship between MEAF and the ICG-PDR variation – rate on this cohort. The definition of EAD, as proposed by Olthoff et al., provides a dichotomous variable which needs a vast cohort to obtain statistical significance and provides less details as compared with MEAF. The combination of these considerations led us to choose the MEAF score as measure outcome. 

 With regards to the time point to measure the MEAF Score, please see the matherials and methods section, lines 103-112, where the complete alghorithm is expressed. The MEAF score was calculated with respect to the traditional formula, which has been validated to consider the max AST / ALT and bilirubin value within the first-three post operative days. Please find our comment to the sample size in the comment below. As suggested by your and the reviewer’s acute observation, we realize that the recipient’s biochemical data reported in table 2 present values from post-operative day one could have mislead the reader. However, the alghorithm was correctly calculated. 

Finally, the aim of applying the ICG-PDR in this study was to monitorate the transition of the graft from the donor to the recipient through the ICG-PDR and to compare it with the currently most reliable predictive score of graft and patient’s prognosis. We hope to have clearly answered to your que

5) I echo reviewer 1, and if ` ICG-PDR testing on liver transplant recipients has been a standard-of-care for 20 years at our institution`, why not increase the donor pool? In addition, if ` All patients were considered with a minimum follow-up of 90 days`, why not patient and graft survival at 90-day?

Reply 5: Dear Editor, thank you for your observations. Although we have used the ICG-PDR after liver transplantation for a long time at our institution, with some recently published articles on the issue (PMID: 32166552 and PMID: 34023468), the practice of measuring the ICG-PDR variation rate between the donor and the recipient is new. However, to explain the relatively small size of our cohort, some aspects need to be acknowledged: during the study period, our Institution had an organ debt with other institutions, thus many of the harvested organs were used elsewhere, whereas many of the organs transplanted at our institution were procured at other centers. Therefore, it was not possible for us to perform a complete assessment of the donor – recipient ICG-PDR on the reported sample. We hope that you may consider our study as a “pilot study”. However, in agreement with you and the reviewers, we are willing, if necessary, to change the study title from “a pilot study” to “case series” – and hope that this would satisfy your request. Please see the new title. 

 Furthermore, to address your last question we highlight that in our cohort the rates of death, primary non function (PNF) and re-transplant were 0% up to 90-days as mentioned in the results, lines 155-156, and in table 3.

 Moreover, to increase the quality of the study, we have followed your and reviewer #1’s suggestion by updating the following data to the materials and methods and results sections: i) the 90-day patient follow-up (expressed as HFD: Hospital Free Days). Please see the implementation within the materials and methods, lines 113-116. 

 In fact the transplant patients with an improved ICG-PDR from the donor to the recipient, expressed as positive log ICG-PDR variation rate, had a higher HFD (mean, 68.67); (SD 6.56) vs patients with a negative log ICG-PDR variation rate – HFD (mean, 62.83); (SD 6.13), although without reaching a statistical significance. These findings support our results, showing that the up or downtrend of the ICG-PDR from the donor-to the recipient translated also in different HFD. Please see the red text in the new version of our manuscript, section results, lines 151-154.

6) Furthermore, indeed, the inclusion of data on `hard` endpoints would increase the quality of the study. 

Consider report patient and graft survival and the occurrence of early allograft dysfunction by conventional criteria (as those proposed by Olthoff et al.). Those would be important considering the intention to propose a predictive test of graft performance and early outcome after transplantation. 

Reply 6: As reported above, we chose the MEAF score since recent evidence points towards it outperforming EAD in terms of patient and graft survival after transplant. Below are the relative references:

- Jochmans I, Fieuws S, Monbaliu D, Pirenne J. "Model for Early Allograft Function" Outperforms "Early Allograft Dysfunction" as a Predictor of Transplant Survival. Transplantation. 2017 Aug;101(8):e258-e264. doi: 10.1097/TP.0000000000001833. PMID: 28557956.

- Pareja E, Cortes M, Hervás D, Mir J, Valdivieso A, Castell JV, Lahoz A. A score model for the continuous grading of early allograft dysfunction severity. Liver Transpl. 2015 Jan;21(1):38-46. doi: 10.1002/lt.23990. Epub 2014 Nov 24. PMID: 25204890.

- Barrueco-Francioni JE, Seller-Pérez G, Lozano-Saéz R, Arias-Verdú MD, Quesada-García G, Herrera-Gutiérrez ME. Early graft dysfunction after liver transplant: Comparison of different diagnostic criteria in a single-center prospective cohort. Med Intensiva (Engl Ed). 2020 Apr;44(3):150-159. English, Spanish. doi: 10.1016/j.medin.2018.09.004. Epub 2018 Oct 24. PMID: 30528954.

As mentioned above, being a continuous grading scale for graft dysfunction, the MEAF score can provide additional, granular information that could be used both clinically and as a surrogate endpoint of transplant survival in clinical trials unlike the categorical values obtained for EAD. 

Furthermore, we found it statistically and methodologically more appropriate to study the potentienl relationship between MEAF and the ICG-PDR variation – rate on this particular cohort. The definition of EAD, as proposed by Olthoff et al., provides a dichotomous variable which needs a vast cohort to find statistical significance. 

- Jochmans I, Fieuws S, Monbaliu D, Pirenne J. "Model for Early Allograft Function" Outperforms "Early Allograft Dysfunction" as a Predictor of Transplant Survival. Transplantation. 2017 Aug;101(8):e258-e264. doi: 10.1097/TP.0000000000001833. PMID: 28557956.

In the previously submitted manuscript we did not configure the transplant outcomes in terms of EAD because as we mentioned above the current evidence from the literature led us to choose a different outcome (MEAF) which is also methodologically appliable to our population. It would certainly be interesting to compare these findings with EAD as outcome, but certainly a large population of liver transplants would be required. 

This is also why we calculated the ICG-PDR variation as logarithm %. We decide to Log our series to straight out exponential growth features and to reduce heteroscedasticity, stabilizing the variance. This elaboration could build data that can be better fitted by a linear model.

The combination of these considerations led us to choose the MEAF as measure outcome.

7) In the Discussion, the authors advocate that the current study supports the hypothesis that ICG-PDR variation rate could represent a measure of the ischemia-reperfusion injury. However, they must admit that limitations of the study population (good quality grafts and low-risk recipients) may compromise this assumption. 

Reply 7: Thank you for your observation. You are right, however, as previously presented by Plevris et al., we share the theory that ICG-PDR could mirror IRI. The IRI-induced injury translates into graft dysfunction that can be measured by the ICG-PDR. Please see the ref below:

- Plevris JN, Jalan R, Bzeizi KI, Dollinger MM, Lee A, Garden OJ, et al. Indocyanine green clearance reflects reperfusion injury following liver transplantation and is an early predictor of graft function. J Hepatol. 1999; 30(1):142-8. 

- Seifalian AM, El-Desoky H, Delpy DT, Davidson BR. Effects of hepatic ischaemia/reperfusion injury in a rabbit model of Indocyanine Green clearance. Clin Sci (Lond). 2002 May;102(5):579-86. PMID: 11980578.

- El-Desoky A, Seifalian AM, Cope M, Delpy DT, Davidson BR. Experimental study of liver dysfunction evaluated by direct indocyanine green clearance using near infrared spectroscopy. Br J Surg. 1999 Aug;86(8):1005-11. doi: 10.1046/j.1365-2168.1999.01186.x. PMID: 10460634.

8) Also, in the Discussion, authors must comment on the feasibility of performing ICG-PDR tests in the donor before entering the operating room for graft retrieval. In many countries, such interventions are not allowed in the donor. Therefore, ethical and legal regulations must limit the applicability of these findings.

Reply 8: ICG-PDR is a non-invasive measurement of liver function performed with the inert dye indocyanine, in which a vial of ICG containing 25 mg of ICG powder is diluted using 5 mL of physiological solution to obtain a concentration of 5 mg/mL (VERDYE 5 mg/mL—Diagnostic Green GmbH, Otto-Hahn-Str. 20, Aschheim-Dornach, Germany). After venous injection, a characteristic curve shows the shape depending on liver metabolism performance, which is exclusively mediated by selective organic anion transporters located in the hepatocyte plasma membrane. ICG is then excreted into the bile without enterohepatic recirculation. We performed the ICG-PDR test totally non-invasively by pulse spectrophotometry using a finger-clip sensor. 

With regards to the ethical and legal regulations in Italy, the test is only performed once the Brain Death certification procedure has been completed by the medical commission (Italian Law n.578/1983), and after checking the willingness of the patients’ relatives to donate the organs and after obtaining their consent to do so (Italian Law n.91/1999); only then is the donor transferred to the operating room for organ retrieval and this procedure can be conceptually compared to coronary angiography. This procedure is approved by the IRB. Following the Editor’s observation we include this ethical and legal limit in the new version of the manuscript, please see the section materials and methods – study protocol, lines 119-121.

 

We note that your study involved tissue/organ transplantation. Please provide the following information regarding tissue/organ donors for transplantation cases analyzed in your study.

1. Please provide the source(s) of the transplanted tissue/organs used in the study, including the institution name and a non-identifying description of the donor(s).

Reply 1: Following your suggestion we have added that organ procurement was carried out at the University Hospital of Udine under the work of the Regional Transplant Center for Friuli Venezia Giulia, by its Head Dr. Roberto Peressutti (one of the authors), as reported in the matherials and methods section, study design, lines 79-81. We also specify that informed consent for organ donation was obtained with respect to the Italian Law, art.91/99 and 578/93. 

 2. Please state in your response letter and ethics statement whether the transplant cases for this study involved any vulnerable populations; for example, tissue/organs from prisoners, subjects with reduced mental capacity due to illness or age, or minors. 

 - If a vulnerable population was used, please describe the population, justify the decision to use tissue/organ donations from this group, and clearly describe what measures were taken in the informed consent procedure to assure protection of the vulnerable group and avoid coercion. 

 - If a vulnerable population was not used, please state in your ethics statement, 

Reply 2: “None of the transplant donors originated from a vulnerable population.” Please see the new version of the manuscript, section materials and methods – study design, line 82.

 3. In the Methods, please provide detailed information about the procedure by which informed consent was obtained from organ/tissue donors or their next of kin. In addition, please provide a blank example of the form used to obtain consent from donors, and an English translation if the original is in a different language.

Reply: Thank you for your question. With regards to the ethical and legal regulations in Italy, the test is only performed once the Brain Death certification procedure has been completed by the medical commission (Italian Law n.578/1983), and after checking the patient’s or the patients’ relatives willingness to donate the organs (Italian Law n.91/1999); only then is the donor transferred to the operating room for organ retrieval and ICG-PDR testing, as coronary angiography (for the cardiac graft evaluation). This procedure has been improved since 2019 as standard-of-care at our Institution and has been approved by the IRB. A blank copy of the informed consent for organ donation is provided within the supplemental material. Please see the materials and methods section, study design, lines 79-81.

 4. Please indicate whether the donors were previously registered as organ donors. If tissues/organs were obtained from deceased donors or cadavers, please provide details as to the donors’ cause(s) of death.

Reply 4: As reported above, understanding the willingness of the patients’ relatives to donate the organs and obtaining their consent (Italian Law n.91/1999) was always performed beforehand by the NITp (Nord Italia Transplant Program). In Italy, consent for organ donation is expressed at the moment of the personal ID card issue and renewal (which is obligatory in Italy). All organs were procured from deceased donors following brain death at the University Hospital of Udine, under the management of the Regional Transplant Network, please see the materials and methods, as requested we add this information accordingly in lines 79-81. A detailed table showing donor characteristics, including the cause of death, is provided in the new version of the manuscript, please see the table 1. 

 5. Please provide the participant recruitment dates and the period during which transplant procedures were done (as month and year). 

Reply 5: The participant recruitment dates and the period during which the transplant procedures were performed all fall into the period: October 2019 – December 2020. Please see the matherials and methods, section study design, lines 80-82. 

 6. Please discuss whether medical costs were covered or other cash payments were provided to the family of the donor. If so, please specify the value of this support (in local currency and equivalent to U.S. dollars).

Reply 6: With respect to the Italian Law and the National Health System all the medical procedures (including the medical support of the donor) are financially covered by the Italian government. The family of the donor has no right to receive any form of payment for organ donation. 

 

Reviewers' comments: 

Reviewer's Responses to Questions

Comments to the Author 

1. Is the manuscript technically sound, and do the data support the conclusions?

Reviewer #1: Yes

Reviewer #2: Partly

2. Has the statistical analysis been performed appropriately and rigorously? 

Reviewer #1: Yes

Reviewer #2: No

3. Have the authors made all data underlying the findings in their manuscript fully available?

Reviewer #1: Yes

Reviewer #2: Yes

4. Is the manuscript presented in an intelligible fashion and written in standard English?

Reviewer #1: Yes

Reviewer #2: Yes

5. Review Comments to the Author

 

Reviewer #1: Cherchi et al. evaluated the variation of ICG-PDR in donor and recipient liver. They have found a significant correlation between MEAF value and ICG-PDR variation.

The paper is well-written and organized. Several papers have reported the usefulness of the ICG test of transplant liver to predict early outcomes. This study is unique in which they analyzed the transition of ICG of the same liver from the donor to the recipient. Here are several comments.

Reply: Thank you! 

1) First of all, the cohort size is too small. To attract more readers, the authors should add more patients or prolong the observation period and evaluate the variation ratio with the actual patient outcome, not with the MEAF.

Reply: : Dear reviewer, thank you for your observations. Although we have used the ICG-PDR after liver transplantation for a long time at our institution, with some recently published articles on the issue (PMID: 32166552 and PMID: 34023468), the practice of measuring the ICG-PDR variation rate between the donor and the recipient is new. However, to explain the limited size of our cohort, the following issues need to be acknowledged: during the study period, our Institution had an organ debt with other institutions, thus many of the harvested organs were used elsewhere, whereas many of the organs transplanted at our institution were procured at other centers. Therefore, it was possible for us to perform a complete assessment of the donor – recipient ICG-PDR on the reported sample. We hope that you may consider our study as a “pilot study”. However, in agreement with you and the reviewers, we are willing, if necessary, to change the study title from “a pilot study” to “case series” – and hope that this would satisfy your request. Please see the new title.

 Furthermore, to increase the quality of the study, we have followed your and the Editor’s suggestion by adding the following data to the materials and methods and results sections: i) a 90-day patient’s follow-up (expressed as HFD: Hospital Free Days). Please see the materials and methods, study protocol, lines 113-116.

 In fact the transplant patients with an improved ICG-PDR from the donor to the recipient, expressed as positive log ICG-PDR variation rate, had a higher HFD (mean, 68.67); (SD 6.56) vs patients with a negative log ICG-PDR variation rate – HFD (mean, 62.83); (SD 6.13), although without reaching a statistical significance. These findings support our results, showing that the up or downtrend of the ICG-PDR from the donor-to the recipient translated also in different HFD. Please see the red text in the new version of our manuscript, results, lines 151-154. 

(We would like to draw your attention to the case of the below mentioned article which was the first to evaluate the thickening fraction in mechanical ventilated patient on a cohort of just seven patients: “Grosu HB, Lee YI, Lee J, Eden E, Eikermann M, Rose KM. Diaphragm muscle thinning in patients who are mechanically ventilated. Chest. 2012 Dec;142(6):1455-1460. doi: 10.1378/chest.11-1638. PMID: 23364680” reach 134 Scopus citation”). We hope that you will embrace publishing this article as a case series despite its limited cohort size considering its novelty and the fact that it is the first research to investigate this relationship. 

2) ICG is now used in other fields other than the hepatology field to evaluate vascularity or blood flow. Please mention this in the introduction section.

Reply: In accordance with your suggestion, we have added this information to the introduction section as follows: “Indocyanine green (ICG) is a cyanine dye safely used in medical diagnostics; although with different techniques, ICG boasts broad applications from vascular and lymphatic perfusion assessments in the fields of gastrointestinal surgery and gynecology, to the vascular and organ functional assessment in hepatology.” Please see the red text in the new version of the manuscript, introduction, lines 54-57.

- Rossi EC, Ivanova A, Boggess JF. Robotically assisted fluorescence-guided lymph node mapping with ICG for gynecologic malignancies: a feasibility study. Gynecol Oncol. 2012 Jan;124(1):78-82. doi: 10.1016/j.ygyno.2011.09.025. Epub 2011 Oct 

- Blanco-Colino R, Espin-Basany E. Intraoperative use of ICG fluorescence imaging to reduce the risk of anastomotic leakage in colorectal surgery: a systematic review and meta-analysis. Tech Coloproctol. 2018 Jan;22(1):15-23. doi: 10.1007/s10151-017-1731-8. Epub 2017 Dec 11. PMID: 29230591.

- Coubeau L, Frezin J, Dehon R, Lerut J, Reding R. Indocyanine green fluoroscopy and liver transplantation: a new technique for the intraoperative assessment of bile duct vascularization. Hepatobiliary Pancreat Dis Int. 2017 Aug 15;16(4):440-442. doi: 10.1016/S1499-3872(17)60040-7. PMID: 28823377.

3) The result of ICG would be affected by cardiac function. Please include the limitation of ICG testing itself.

Reply: In agreement with the reviewer, we have included this important suggestion in the discussion section, final paragraph – limitations, lines 267-269. Thank you!

4) Since this is a retrospective study, outcomes of the LT will be available. Although the authors used MEAF as a surrogate marker for the outcome, is there any correlation with a real outcome?

Reply: As reported above, recent evidence points towards the MEAF score outperforming EAD in terms of patient and graft survival after transplant, thus providing the all-important support for its use in this research. The relative references are as follows:

- Jochmans I, Fieuws S, Monbaliu D, Pirenne J. "Model for Early Allograft Function" Outperforms "Early Allograft Dysfunction" as a Predictor of Transplant Survival. Transplantation. 2017 Aug;101(8):e258-e264. doi: 10.1097/TP.0000000000001833. PMID: 28557956.

- Pareja E, Cortes M, Hervás D, Mir J, Valdivieso A, Castell JV, Lahoz A. A score model for the continuous grading of early allograft dysfunction severity. Liver Transpl. 2015 Jan;21(1):38-46. doi: 10.1002/lt.23990. Epub 2014 Nov 24. PMID: 25204890.

- Barrueco-Francioni JE, Seller-Pérez G, Lozano-Saéz R, Arias-Verdú MD, Quesada-García G, Herrera-Gutiérrez ME. Early graft dysfunction after liver transplant: Comparison of different diagnostic criteria in a single-center prospective cohort. Med Intensiva (Engl Ed). 2020 Apr;44(3):150-159. English, Spanish. doi: 10.1016/j.medin.2018.09.004. Epub 2018 Oct 24. PMID: 30528954.

Being a continuous grading scale for graft dysfunction, the MEAF score can provide additional, granular information that could be used both clinically and as a surrogate endpoint of transplant survival in clinical trials, unlike the categorical values obtained for EAD. 

Furthermore, we found it statistically and methodologically more appropriate to study the potentienl relationship between MEAF and the ICG-PDR variation – rate on this particular cohort. The definition of EAD, as proposed by Olthoff et al., provides a dichotomous variable which needs a vast cohort to obtain statistical significance. Instead, a continuous variable – expecially if elaborated by log transformation to build a linear model – could describe the association between the analyzed variables better. The combination of these considerations led us to choose the MEAF score as measure outcome. 

Following your and the Editor’s suggestion to increase the quality of the study, a 90 days patient’s follow-up has been updated in the results. The Hospital-Free survival Days (HFD) summarizes the postoperative outcome as the number of days patients spent outside of any healthcare facility in an observation period of 90 days after OLTx. As per your request, there is also a correlation with the real outcome. Since the 90-days mortality, PNF and re-transplant endpoints were 0%, we summarized the real outcome by means of HFD: the transplant patients with an improved ICG-PDR from the donor to the recipient, expressed as positive log ICG-PDR variation rate, had a higher HFD (mean, 68.67); (SD 6.56) vs patients with a negative log ICG-PDR variation rate – HFD (mean, 62.83); (SD 6.13), although without reaching a statistical significance. These findings support our results, showing that the up or downtrend of the ICG-PDR from the donor-to the recipient translated also in different postoperative hospitalization. Please see the red text in the materials and methods and results sections.

5) T1 ICG was obtained from DBD, and the liver should be normal. Why was T1 high in some patients?

Reply: Thank you for your observation. The high T1 values in some patients was, in fact, the reason why we calculated the ICG-PDR variation as a logarithm %. We decided to Log our series in order to straighten out the exponential growth features and to reduce heteroscedasticity, stabilizing the variance. This transformation could produce data that might be fitted better by a linear model.

Many studies in the literature show that high ICG-PDR values are characteristic of/correlate with good quality grafts. That said, no “clear” definition of a “good quality graft” by means of ICG-PDR has yet been established, and further studies are needed. At the moment, a “good” ICG-PDR does not represent a selection criteria for graft suitability for transplant. Liver biopsy continues to be the gold standard for selecting graft eligibility for transplant. 

6) DO the authors think ICG-PDR is a real-time test to assess graft dysfunction or a predictive marker for future graft dysfunction?

Reply: Thank you for your cogent inquiry. Yes, we are convinced that ICG-PDR is a real-time test that can be used for assessing graft function non invasively or as a predictive marker for future graft dysfunction. Please see our recent publications on ICG-PDR and it’s value for short and long term prognosis after OLTx. Please see: PMID: 34023468; PMID: 32166552.

 

Reviewer #2: Dear Author 

Congratulations on your study. 

I have some concerns about and I would like to discuss with you. 

1) If you have since 2019, ICG-PDR implemented as standard-of-care for all livers donors and transplant recipientes why you don't try increased your cohort of patients? How many liver transplants do you perform in your center in a year? Maybe with this analysis will be possible have more consistent and definitive results.

Reply: Dear reviewer, thank you for your observations. It is true that we have used the ICG-PDR after liver transplantation for a long time at our institution, with some recently published articles: PMID: 32166552 and PMID: 34023468. However, the practice of measuring the ICG-PDR variation rate between the donor and the recipient is a new. 

 Moreover, to explain the limited size of our cohort, the following issues need to be acknowledged: during the study period, our Institution had an organ debt with other institutions, thus many of the harvested organs were used elsewhere, whereas many of the organs transplanted at our institution were procured at other centers. Therefore, it was possible for us to perform a complete assessment of the donor – recipient ICG-PDR on the reported sample. We hope that you may consider our study as a “pilot study”. However, in agreement with you, the Editor and the other reviewers, we are willing, if necessary, to change the study title from “a pilot study” to “case series” – and hope that this would satisfy your request. Please see the new title. Our center is a medium transplant center with about 30–35 liver, 25–30 heart transplants and 60–70 kidney transplants performed each year. 

 Furthermore, to increase the quality of the study, we have followed your and the Editor’s suggestion by adding the following data to the materials and methods and results sections: a 90-day patient’s follow-up (expressed as HFD: Hospital Free Days). Please see the materials and methods, study protocol, lines 113-116.

In fact, the transplant patients with an improved ICG-PDR from the donor to the recipient, expressed as positive log ICG-PDR variation rate, had a higher HFD (mean, 68.67); (SD 6.56) vs patients with a negative log ICG-PDR variation rate – HFD (mean, 62.83); (SD 6.13), although without reaching a statistical significance. These findings support our results, showing that the up or downtrend of the ICG-PDR from the donor-to the recipient translated also in different postoperative hospitalization. Please see the red text in the results section, lines 151-154.

(We would like to draw your attention to the case of the below mentioned article which was the first to valuate the thickening fraction in mechanical ventilated patient on a cohort of just seven patients: “Grosu HB, Lee YI, Lee J, Eden E, Eikermann M, Rose KM. Diaphragm muscle thinning in patients who are mechanically ventilated. Chest. 2012 Dec;142(6):1455-1460. doi: 10.1378/chest.11-1638. PMID: 23364680” reach 134 Scopus citation”). We hope that you will embrace publishing this article as a pilot study despite its limited cohort size considering its novelty and the fact that it is the first paper to investigate this relationship. 

2) You put MELD score in table 1. Donor demographical and laboratory data. Please consider to change to Table 2. 

Reply: Thank you for your suggestion, we have moved the data accordingly. Please see the new table 2.

3) Why did you prefer to use the MEAF grade instead other score like the definition proposed by Olthoff et al. ? And why did you use the MEAF score at day one? instead day 3 ?

This could minimizes the contribution of preoperative elevated bilirubin and significant preoperative coagulopathy found in the recipients. 

Reply: We thank you for your question because thanks to this comment we have realized that our content can be misunderstood. The MEAF score was calculated using the traditional formula, which has been validated to consider the max AST and ALT value within the first three post operative days. As suggested by your acute observation, we realize that the recipient’s biochemical data reported in table 2 showing values from post-operative day one may mislead the reader. However, the alghorithm was correctly calculated. 

 We chose the MEAF score since recent evidence points towards it outperforming EAD in terms of patient and graft survival after transplant. Please find the relative references below:

- Jochmans I, Fieuws S, Monbaliu D, Pirenne J. "Model for Early Allograft Function" Outperforms "Early Allograft Dysfunction" as a Predictor of Transplant Survival. Transplantation. 2017 Aug;101(8):e258-e264. doi: 10.1097/TP.0000000000001833. PMID: 28557956.

- Pareja E, Cortes M, Hervás D, Mir J, Valdivieso A, Castell JV, Lahoz A. A score model for the continuous grading of early allograft dysfunction severity. Liver Transpl. 2015 Jan;21(1):38-46. doi: 10.1002/lt.23990. Epub 2014 Nov 24. PMID: 25204890.

- Barrueco-Francioni JE, Seller-Pérez G, Lozano-Saéz R, Arias-Verdú MD, Quesada-García G, Herrera-Gutiérrez ME. Early graft dysfunction after liver transplant: Comparison of different diagnostic criteria in a single-center prospective cohort. Med Intensiva (Engl Ed). 2020 Apr;44(3):150-159. English, Spanish. doi: 10.1016/j.medin.2018.09.004. Epub 2018 Oct 24. PMID: 30528954.

Being a continuous grading scale for graft dysfunction, the MEAF score can provide additional, granular information that could be used both clinically and as a surrogate endpoint of transplant survival in clinical trials unlike the categorical values obtained for EAD. Furthermore, we found it statistically and methodologically more appropriate to study the potentienl relationship between MEAF and the ICG-PDR variation – rate on this particular cohort. The definition of EAD, as proposed by Olthoff et al., provides a dichotomous variable which needs a vast cohort to obtain statistical significance. The combination of these considerations led us to choose the MEAF score as measure outcome. 

4) When you discuss data from donor like steatosis, warm and cold ischemic times, for example, you did not correlate with MEAF, only because the good condition of the donor himself ? Did you considered that could be because you performed the score to soon after liver transplant, on day 1? Or maybe because you had only patients with low MELD and good donors conditions? I would like to know your opinion about.

Reply: Thank you for your question. In order to provide an adequate answer, we have prepared a new table showing the relationship between the MEAF Score and the mentioned variables. Please see the table 4 in the new version of the manuscript. 

 In response to your query about the time point for measuring the MEAF Score, please see the matherials and methods section, where the alghorithm is expressed, lines 103-112. The MEAF score was calculated using the traditional formula, which was validated to consider the max AST and ALT value within the first three post operative days. As suggested by your acute observation, we realize that the recipient’s biochemical data reported in table 2 present values from post-operative day one could have mislead the reader. However, the alghorithm was correctly calculated.

5) The value of ICG-PDR on the first day after transplant is a good predictor for EAD, especially to livers from extended criteria, but you did not use those kind of donors, so the variation rate of ICG "represent a measure of the ischemia and reperfusion injury", like you hypothetically said. So this was not related with donor qualities. Do you agree ?

Reply: Yes we agree with you, also about the fact that ICG-PDR is a good predictor for EAD and MEAF and that it represents a measure of ischemia and reperfusion injury. In fact, the ICG-PDR variation rate is not applied in this context to assess the graft suitability for transplant (or to define it as good or bad); thank you for your observation.

Please see the ref below:

- Plevris JN, Jalan R, Bzeizi KI, Dollinger MM, Lee A, Garden OJ, et al. Indocyanine green clearance reflects reperfusion injury following liver transplantation and is an early predictor of graft function. J Hepatol. 1999; 30(1):142-8. 

- Seifalian AM, El-Desoky H, Delpy DT, Davidson BR. Effects of hepatic ischaemia/reperfusion injury in a rabbit model of Indocyanine Green clearance. Clin Sci (Lond). 2002 May;102(5):579-86. PMID: 11980578.

- El-Desoky A, Seifalian AM, Cope M, Delpy DT, Davidson BR. Experimental study of liver dysfunction evaluated by direct indocyanine green clearance using near infrared spectroscopy. Br J Surg. 1999 Aug;86(8):1005-11. doi: 10.1046/j.1365-2168.1999.01186.x. PMID: 10460634.

We would need to gather data from a multicenter study to solve this issue; indeed, we are in the process of proposing such an investigation at this moment. 

6) You have not seen your data any early allograft dysfunction (EAD), you use only good quality donors and recipients with low MELD. Possibily if our data was more extensive you coud find diferente results? Please I would like to hear your opinion about.

Reply: Thank you for your observation. In the previously submitted manuscript we did not configure the transplant outcomes in terms of EAD because as we mentioned above the current evidence from the literature led us to choose a different outcome (MEAF) which is also methodologically more appropriate to be applied in our population. It would certainly be interesting to compare these findings with EAD as outcome, but certainly a large population of liver transplants would be required. Please see the response to your comment nr. 3. This is also why we calculated the ICG-PDR variation as logarithm %. We decide to Log our series to straight out exponential growth features and to reduce heteroscedasticity, stabilizing the variance. This elaboration could build data that can be better fitted by a linear model.

We feel also to specify that we did not use only good quality donors - our population was equally distributed and did not suffer a selection bias that could influence our results as you correctly mentioned. 

Furthermore, to extensively address your question, the functionality of the liver and consequently the PDR value are strictly dependent on the general clinical conditions (e.g. including the stress from the transplant process) of the patient, being either a donor or recipient. Further studies are needed to define whether there is a statistically significant variation in PDR values between standard and marginal (ECD) donors, but this was not the aim of the study in our case. 

In closing, we hope that our modifications have addressed all the issues raised in your informative and thorough review of the paper. We would be glad to make further changes upon request. On behalf of all the authors, we extend our gratitude to you for your time and assistance with our review. 

We look forward to hearing your response. 

Sincerely yours, 

On behalf of all the authors, 

Marco Ventin, M.D., and Prof. Luigi Vetrugno, M.D.

---

## [Decision Letter · Decision Letter 1]

16 Aug 2021

ASSOCIATION BETWEEN THE DONOR TO RECIPIENT ICG-PDR VARIATION RATE AND THE FUNCTIONAL RECOVERY OF THE GRAFT AFTER ORTHOTOPIC LIVER TRANSPLANTATION: A CASE SERIES

PONE-D-21-16544R1

Dear Dr. Ventin,

We’re pleased to inform you that your manuscript has been judged scientifically suitable for publication and will be formally accepted for publication once it meets all outstanding technical requirements.

Kind regards,

Yuri Longatto Boteon, M.D., Ph.D.

Academic Editor

PLOS ONE

Additional Editor Comments (optional):

I have read the revised version of the manuscript ‘Association between the donor to recipient ICG-PDR variation rate and the functional recovery of the graft after orthotopic liver transplantation: a case series’ with great interest.

Improvements have been made to the manuscript, and the authors addressed significant methodological comments. Due to the predominant descriptive design of the study, I do believe the amendment in the title to ‘A CASE SERIES’ is adequate.

The study data has been made available, and there are no competing interests that may affect the integrity of the review process.

The limited population of the study leads to some limitations, which are better described in the current version of the manuscript. However, I strongly suggest authors pursue Reviewer 2 suggestions, increase patient population and extend the postoperative follow-up period for future studies. Nevertheless, considering the novelty of the study and the fact that it fulfills PLOS ONE publication criteria, I do believe it is acceptable for publication in the current format. 

Reviewers' comments:

Reviewer's Responses to Questions

**Comments to the Author**

1. If the authors have adequately addressed your comments raised in a previous round of review and you feel that this manuscript is now acceptable for publication, you may indicate that here to bypass the “Comments to the Author” section, enter your conflict of interest statement in the “Confidential to Editor” section, and submit your "Accept" recommendation.

Reviewer #1: All comments have been addressed

Reviewer #2: (No Response)

2. Is the manuscript technically sound, and do the data support the conclusions?

Reviewer #1: Yes

Reviewer #2: Partly

3. Has the statistical analysis been performed appropriately and rigorously? 

Reviewer #1: Yes

Reviewer #2: Yes

4. Have the authors made all data underlying the findings in their manuscript fully available?

Reviewer #1: Yes

Reviewer #2: Yes

5. Is the manuscript presented in an intelligible fashion and written in standard English?

Reviewer #1: Yes

Reviewer #2: Yes

6. Review Comments to the Author

Reviewer #1: The authors have fulfilled each of the major compulsory revisions and modified the manuscript as requested. I would think the previous title of “a pilot study” is fine. The paper is not just a report of case series but a study to evaluate the usefulness of ICG-PDR measurement.

Reviewer #2: Dear Author

Thank you again for resubmitting your manuscript and prompt reply the questions.

I think the change the study title to ''case series" cannot help to better understanding the small number of patients analyzed.

Your intention of photographing the liver function in two moments is very interesting for clinical practice indeed. Because of this , in my opinion you could consider to increase your cohort, and add some observation period of time, this way you could avoid some limitations like you comment in discussion: "the small cohort of patients did not allow us to investigate certain aspects."Another issue is about the fact that ICG-PDR is a good predictor for EAD, specially to livers from extended criteria, again maybe if you would gather a bigger data you could solve this issue.

7. PLOS authors have the option to publish the peer review history of their article (what does this mean?). If published, this will include your full peer review and any attached files.

Reviewer #1: No

Reviewer #2: No

---

## [Editor Report · Acceptance letter]

20 Aug 2021

PONE-D-21-16544R1 

ASSOCIATION BETWEEN THE DONOR TO RECIPIENT ICG-PDR VARIATION RATE AND THE FUNCTIONAL RECOVERY OF THE GRAFT AFTER ORTHOTOPIC LIVER TRANSPLANTATION: A CASE SERIES 

Dear Dr. Ventin:

I'm pleased to inform you that your manuscript has been deemed suitable for publication in PLOS ONE. Congratulations! Your manuscript is now with our production department. 

Kind regards, 

on behalf of

Prof. Yuri Longatto Boteon 

Academic Editor

PLOS ONE